# Cardiopulmonary Ultrasound Patterns of Transient Acute Respiratory Distress of the Newborn: A Retrospective Pilot Study

**DOI:** 10.3390/children10020289

**Published:** 2023-02-02

**Authors:** Maria Pierro, Roberto Chioma, Consuelo Benincasa, Giacomo Gagliardi, Lorenzo Amabili, Francesca Lelli, Giovanni De Luca, Enrico Storti

**Affiliations:** 1Neonatal and Paediatric Intensive Care Unit, M. Bufalini Hospital, AUSL Romagna, 47521 Cesena, Italy; 2Neonatal Intensive Care Unit, Catholic University of Sacred Heart, 00168 Rome, Italy; 3Paediatric Unit, Ferrara University Hospital, 44124 Ferrara, Italy; 4Data Science Department, Mosaic Factor, 08018 Barcelona, Spain; 5Pathology and Oncohematology Department, M. Bufalini Hospital, AUSL Romagna, 47521 Cesena, Italy; 6Department of Critical Care, Maggiore Hospital, 26100 Cremona, Italy

**Keywords:** transient tachypnea of the newborn, lung ultrasound, targeted neonatal echocardiography, meconium aspiration syndrome, persistent pulmonary hypertension of the newborn, phenotype

## Abstract

Acute transient respiratory distress in the first hours of life is usually defined as transient tachypnea of the newborn (TTN). TTN is a respiratory self-limiting disorder consequent to delayed lung fluid clearance at birth. While TTN is the most common etiology of respiratory distress near term, its pathogenesis and diagnostic criteria are not well-defined. Lung ultrasound and targeted neonatal echocardiography are increasingly being used to assess critically ill infants, although their combined use to improve diagnostic precision in neonatal intensive care units has not yet been described. This retrospective pilot analysis aimed to identify possible cardiopulmonary ultrasound (CPUS) patterns in term and late preterm infants suffering from transient respiratory distress and requiring non-invasive respiratory support. After retrospectively revising CPUS images, we found seven potential sonographic phenotypes of acute neonatal respiratory distress. Up to 50% of the patients presented with signs of increased pulmonary vascular resistance, suggesting that those patients may be diagnosed with mild forms of persistent pulmonary hypertension of the newborn. Approximately 80% of the infants with a history of meconium-stained amniotic fluid displayed irregular atelectasis, indicating that they may have suffered from mild meconium aspiration syndrome. CPUS evaluation may improve accuracy in the approach to the infants presenting with transient acute respiratory distress, supporting communication with the parents and carrying important epidemiological consequences.

## 1. Introduction

Acute, self-limiting respiratory distress after birth is usually diagnosed as transient tachypnea of the newborn (TTN). TTN is a respiratory disorder that begins within the first few hours of life, as a consequence of an altered adaptation to extrauterine life [1]. Although TTN is a benign disease, it often requires admission to the Neonatal Intensive Care Unit (NICU) to provide respiratory support, causing considerable parental stress and a significant contribution to healthcare costs [2,3]. TTN definition is vague and represents the description of a symptom, rather than a physiopathological entity(s). TTN is usually diagnosed retrospectively in those infants that develop benign, self-limiting respiratory distress in the first hours of life, characterized by a cluster of clinical signs, including tachypnea, expiratory braking, grunting, and poor oxygenation, which resolve with oxygen administration and/or non-invasive respiratory support [4]. The need for mechanical ventilation usually suggests a different cause of respiratory distress or the development of a complication. The suggested aetiology underlying TTN is a delayed clearance of fetal lung fluid, resulting mainly from an altered activation of the epithelial sodium channels in the lung during labor and delivery [1]. Therefore, TTN is also called wet lung [5]. The persistence of fetal cardiopulmonary circulation with increased pulmonary vascular resistance (PVR) has been sporadically described in patients eventually diagnosed with TTN [6]. However, the exact incidence of altered PVR in patients with acute transient respiratory distress in the first few hours of life has not been fully investigated. Chest X-ray is often performed when suspecting TTN mainly to exclude other more serious respiratory conditions, as no pathognomonic signs of TTN can be radiologically detected [1]. Chest X-ray not only has limitations but also poses neonatal tissues and organs at risk of latent radiation effects [7]. 

LUS is emerging as an important tool in the assessment of neonatal respiratory disorders, including TTN [8]. Targeted neonatal echocardiography (TnEcho) has long been used in the NICU to evaluate cardiac functional aspects. However, the systematic use of TnEcho in the assessment of the pulmonary vascular status in patients with transient respiratory distress after birth has never been proposed.

This retrospective pilot analysis aimed to evaluate, through a cardiopulmonary ultrasound (CPUS), the presence of different sonographic phenotypes in term and late-preterm infants presenting with transient respiratory distress and requiring non-invasive respiratory support in the first few hours after birth. 

## 2. Materials and Methods

### 2.1. Patients, Equipment, Study Design

This retrospective pilot study included all the patients born at 34 weeks gestational age or older, requiring non-invasive respiratory support, who had undergone pulmonary ultrasound and TnEcho in the first 24 h of life. To be included in the analysis, patients had to be on respiratory assistance at the time of CPUS examination. Data were collected from patients admitted to the Neonatal and Pediatric Intensive Care Unit of Bufalini Hospital (Cesena, Italy) between July 2020 to July 2022. During hospital stay, CPUS evaluation was performed as a non-invasive radiation-free follow-up study and not with the intent of establishing a differential diagnosis. The discharge diagnosis was based on the more established clinical and radiological criteria.

Clinical data were collected retrospectively from the computerized and paper medical records, de-identified, and reported in an Excel database. Maternal history was reviewed and recorded. Definitions are detailed in the Appendix A. Briefly, diabetes in pregnancy was defined according to NICE guidelines [9,10]; gestational disorders of pregnancy were classified according to the American College of Obstetricians and Gynecologists (ACOG) guidelines [11]; “Intrauterine Inflammation or Infection or both” also known as “Triple-I syndrome” was defined according to Peng et al. [12], intrauterine growth restriction (IUGR) was defined as a fetal body weight below the 10th percentile at fetal ultrasound [13], fetal heart monitoring was graded in the three categories of severity [14].

Perinatal infant data (gestational age, birth weight, need for resuscitation in the delivery room, Apgar score, and the presence of stained amniotic fluid), infant data concerning NICU stay, including data related to potential infections (inflammation indices, blood culture, and use of antibiotics), neonatal short-term outcomes (length of stay, duration and type of non-invasive respiratory support, maximum fraction of inspired oxygen (FiO_2_), and need for surfactant administration), sonographic examinations (pulmonary and cardiac ultrasounds findings), clinical data at the moment of the examinations (FiO_2_, hours of life), neonatal vital parameters (heart rate and blood pressure) were collected for each newborn.

Cardiac and pulmonary ultrasound evaluation was performed by an experienced neonatologist using a high-frequency (9–16 MHz) linear probe with a General Electric Medical System LOGIQ S8 ultrasound machine. Off-line assessment of the collected sonographic images was performed by an independent operator trained in neonatal CPUS and blinded to the clinical characteristics of the patients.

#### 2.1.1. Lung Ultrasound (LUS) Data

LUS was systematically performed on all lung fields: Anterior superior, lateral superior, posterior superior, anterior inferior, lateral inferior, and posterior inferior for the right and left sides. The LUS assessment was based on the analysis of the standard artifacts:–A-lines that have horizontal lines arising at regular intervals from the pleural line (Figure 1A). A-lines are reverberating artifacts due to reflections of the ultrasound beam from the normal pleura overlying a normally aerated lung [14]–B-lines are long, vertical, well-defined, hyperechoic, dynamic lines originating from the pleural line (Figure 1B). The pleura, covering a hyperdense non-consolidated lung, shows a variable number of irregularities and acoustic traps on its surface, generating B-lines at LUS [15];

The interpretation of LUS artifacts and images generates various LUS patterns.


A- Pattern with normal lung sliding. The A-pattern is characterized by the presence of A-lines and less than 3 isolated B-lines (Figure 1A). This pattern describes the normal lung. The presence of lung sliding, which is the normal movement of the visceral pleura against the motionless parietal pleura, proves the absence of pneumothorax.B1-pattern. The B1-pattern consists of three or more non-confluent B-lines per scan (Figure 1B) [16,17,18]. According to the available evidence [19], we considered this pattern normal lung for newborn infants.Double lung point. The double lung point represents a sharp sonographic demarcation between the upper and lower lung fields, with less compact B-lines in the former than in the latter, suggesting a gravity-dependent pattern (Figure 1C) [17]. The presence of the double lung point suggests increased fluid in the interstitial space, due to a decreased clearance of pulmonary fluid during labor and delivery (wet lung) [20].B2-pattern. The B2-pattern consists in the confluence of B-lines that occupy the entire intercostal space between two ribs, suggesting a further increase in the interstitial fluid with a gravity-dependent pattern. Pleural line is normal (Figure 1D) [16]. According to the literature, this was still interpreted as a sign of wet lung [21,22].White lung with irregular pleural line. The white lung is characterized by compact B-lines that cause the acoustic shadow of the ribs to disappear within the entire scanning zone, anteriorly and posteriorly without spared areas, with thickened and irregular pleural line (Figure 1E). This pattern is usually accompanied by the ground-glass opacity (GOS) sign, characterized by mild, regularly distributed lung consolidations with no obvious air bronchogram (Figure 1E), or by the snowflake (SFS) sign [21,22], characterized by regularly distributed lung consolidations with air bronchogram that resembles a snow pattern [23,24]. This pattern is typical of the respiratory distress syndrome (RDS), which is caused by a dysfunction or lack of lung surfactant [21,22].Irregular atelectasis. This pattern is characterized by the presence of lung consolidations with irregular margins, along with a few spared areas [25,26]. The presence of atelectasis is characterized on LUS by tissue-like images with anechogenic borders with or without air bronchogram [27]. The presence of the atelectasis is irregularly distributed in the lung, may be more evident on one side, and does not follow a gravity-dependent pattern (Figure 1F-H) [24]. We defined this pattern as pulmonary consolidation.


In addition to the qualitative description of the LUS patterns, we assigned a score in order to standardize the establishment of the LUS diagnosis and statistically compare the various lung fields, which are often different from each other. Score 0 was assigned to the A-pattern, score 1 to the B1-pattern, score 2 to the double lung point, score 3 to the B2-pattern, score 4 to white lung with irregular pleural line +/− GOS or SFS, and score 5 in the case of irregular atelectasis (Table 1). If the lung was not homogeneous (different patterns/scores in the different lung fields), the higher score in any lung field was selected to establish the LUS diagnosis.

Based on the diagnostic score, there were 4 possible LUS diagnoses: (i) Normal lung (score 0–1), (ii) wet lung, suggesting an altered clearance of fetal lung fluid at birth (score 2–3), (iii) RDS, suggesting a surfactant deficiency (score 4), and (iv) atelectasis, suggesting the aspiration of stained amniotic fluid (score 5) (Table 1, Figure 1). 

#### 2.1.2. Echocardiographic Data

The main goal of TnEcho in our study was to evaluate the presence of increased PVR. We considered that pulmonary resistances were increased if one of the following was present [28]:Left ventricle telesystolic eccentricity index (EI) > 1.15 (Figure 2A). EI was obtained from the parasternal short axis at the mid-papillary muscle level. The formula (EI = D2/D1) was used, where D1 is the ventricular diameter perpendicular to the interventricular septum bisecting D2, the diameter parallel to the interventricular septum [28]. As the right-to-left ventricular pressure ratio increases, septal curvature typically flattens and may even reverse curvature, providing higher EI.Pulmonary artery pressure-systolic (PAPs) systemic or supra-systemic (Figure 2B). PAPs estimation was based on Doppler measurement of tricuspid regurgitation and pulmonary regurgitation jet on parasternal short axis view. The velocity at end-diastole (at the QRS complex on the ECG) is converted into a pressure gradient through the modified Bernoulli equation: (RVSP = 4 × v2), where v is the maximum velocity of the tricuspid valve regurgitation jet measured using continuous wave (CW) Doppler. Right atrial pressure was ignored.Pulmonary artery acceleration time to right ventricular ejection time ratio (PAAT/RVET) < 0.3 (Figure 2C) +/− pulmonary notch (Figure 2D). PAAT/RVET was taken on the parasternal short axis view or parasternal long axis [29]. Pulmonary artery acceleration time (PAAT) is defined as the interval between the onset of systolic pulmonary arterial flow and peak flow velocity. Right ventricle ejection time (RVET) is the interval between the onset of right ventricle ejection to the point of systolic pulmonary arterial flow cessation. PAAT may be shortened in the case of increased PVR for several reasons: enhanced early pulmonary ejection, increased pulmonary vascular resistance, and loss of lung compliance leading to a rapid increase and reduction of flow velocity. PAAT, in fact, represents pulmonary flow acceleration, which increases as the vascular resistance is augmented, based on Newton’s law of motion. Therefore, the time to peak velocity in the pulmonary artery decreases and PAAT shortens, while the ejection time stays the same, causing a decreased PAAT/RVET ratio.PDA pattern with bidirectional pattern and right-to-left shunt more than 30%, or continuous right-to-left pattern [30] (Figure 2E).

#### 2.1.3. Placental Histology

We retrospectively evaluated placental histology. The presence of maternal vascular malperfusion (MVM) and chorioamnionitis was defined according to the Amsterdam International Consensus criteria [31]. Representative pictures are shown in Figure 3. A detailed description of placental histology definitions is reported in the Appendix A.

### 2.2. Statistical Analysis

Univariate analyses were used to evaluate the association of the acute neonatal distress (AND) phenotypes (Table 2) with the duration of ventilation, duration of oxygen therapy, length of hospitalization, and prenatal and perinatal characteristics. We also studied the association of increased PVR presence, as well as LUS diagnosis, with the same variables. Descriptive data are presented using mean and standard deviation for continuous variables, median and interquartile ranges if the data are not normally distributed, and proportions for dichotomous variables. We compared continuous variables using ANOVA (means) or Wilcoxon Rank-Sum (medians) and dichotomous variables using a chi-square test or Fisher’s exact test if the expected cell frequencies were less than five. All the tests were two-tailed and *p* < 0.05 was considered statistically significant. In the case of a significant result in the ANOVA, a post-hoc test (Bonferroni) was carried out in order to detect which groups showed a significant difference from each other. All images were evaluated in duplicate, and we used Cohen’s kappa (κ) with 95% confidence intervals to calculate the adjusted chance agreement between operators. Statistical analyses were conducted using R, version 14 (StataCorp LP, College Station, TX, USA).

## 3. Results

During the study period, 24 patients met the inclusion criteria and were enrolled in this retrospective pilot analysis. The data of the general study population are reported in Table 2. The number of excluded patients and the reasons for exclusion are reported in Appendix A.

### 3.1. LUS and Echo Findings

We found that overall, 50% of the included patients (*n* = 12) displayed signs of increased PVR at TnECHO. Out of these, four patients (16.6% of the entire population) had a normal LUS, therefore impaired pulmonary circulation was the only explanation for respiratory distress and NICU admission. In the remaining eight cases of increased PVR (33.3% of the entire population), LUS showed either wet lung, RDS or consolidations (Table 3).

In total, eight patients (33.3%) presented with wet lung at LUS. Half of these infants (*n* = 4, 16.6% of the entire population) showed signs of increased PVR at TnEcho. Irregular atelectasis was detected in approximately 33.3% of the patients (*n* = 8), of which two patients displayed increased PVR (8.3% of the entire population). Two patients were diagnosed with RDS at LUS and they both had increased PVR. Finally, 8.3% of the patients showed a normal LUS and echo (*n* = 2). The agreement between operators was high, as κ (with 95% confidence intervals) was 0.90 (0.87–0.92) for the LUS classification, while it was one (no disagreements) for the TnEcho classification, intended as the presence or absence of increased PVR. 

### 3.2. Acute Neonatal Distress (AND) Phenotypes

According to the CPUS findings, in our study population, we found seven different types of respiratory distress, which we named AND phenotypes (Table 3 and Table 4).


Undefined phenotype (AND-u) (*n* = 2, 8.3%): normal LUS with TnECHO findings indicative of normal PVR;Vascular phenotype (AND-v) (*n* = 4, 16.6%): normal LUS with TnECHO findings indicative of increased PVR;Wet lung phenotype (AND-w) (*n* = 4, 16.6%): LUS signs of increased lung fluid with TnECHO findings indicative of normal PVR;Vascular-wet lung phenotype (AND-vw) (*n* = 4, 16.6%): LUS signs of increased lung fluid with TnECHO findings indicative of increased PVR;Vascualar-RDS (*n* = 2, 8.3%): LUS signs of RDS with TnECHO findings indicative of increased PVR.Consolidation Phenotype (AND-c) (*n* = 6, 25%): irregular lung consolidation at LUS with TnECHO findings indicative of normal PVR;Vascular-consolidation Phenotype (AND-vc) (*n* = 2, 8.3%): irregular lung consolidation at LUS with TnECHO findings indicative of increased PVR;


### 3.3. AND Phenotypes, Perinatal Features, and Neonatal Outcomes

Although the study was not powered to detect significance regarding the different AND phenotypes and prenatal conditions or neonatal outcomes, we ran preliminary statistics in order to eventually assist and adequately power further prospective studies. 

The strongest association that we found was between stained amniotic fluid and the presence of lung atelectasis at LUS. All the infants showing irregular consolidations had stained amniotic fluid (mostly meconium stained and in one case blood stained) (Appendix A).

We found a trend towards an increased incidence of MVM on placental histology in patients showing increased PVR compared to those who did not show increased PVR (66% vs. 33% *p* = 0.071). In patients showing increased PVR, there was also a trend toward an increased incidence of placental vasculitis (50% vs. 25% *p* = 0.075). Systolic blood pressure was significantly lower at three hours (42+/−4.2 vs. 49+/−4 mmHg, *p* = 0.026) and five hours of life (43.3+/−2 vs. 53+/−6.5 mmHg, *p* = 0.02) in the case of increased PVR (Appendix A). Ultimately, no patient was diagnosed with systemic hypotension. No other variables showed either a trend or a statistical significance with the presence of increased PVR (Appendix A).

The different LUS diagnoses did not show a difference in the prenatal characteristics or the postnatal outcomes (Appendix A).

### 3.4. CPUS Diagnosis versus Clinical Discharge Diagnosis

The clinical discharge diagnosis of our patients was TTN in 91% of the cases (22 infants), RDS in 4.5% (1 patient), and MAS in 4.5% (one patient) (Table 1). When revising the CPUS images, we found that only a minority of the patients (four cases, 16%) could be diagnosed with the typical TTN, intended as wet lung. 

CPUS evaluation suggests that among infants diagnosed with TTN, there are mild forms of persistent pulmonary hypertension of the newborn (PPHN) and meconium aspiration syndrome (MAS). The proposed CPUS diagnosis of mild PPHN was based on the presence of a pure vascular component without parenchymal involvement (AND-v phenotype). The proposed CPUS diagnosis of MAS was based on the presence of a compatible history (stained amniotic fluid) and typical LUS pattern, characterized by the presence of lung consolidations with irregular margins, along with spared areas (AND-c phenotype).

According to this view, 25% of the patients (six cases) could be classified as mild isolated MAS, 16% as mild isolated PPHN (four cases), 16.6% as isolated TTN (four cases), while 8% (two cases) had an undefined phenotype with normal CPUS examination. The remaining eight infants (35%) suffered from mixed phenotypes, with increased PVR differently combined with the four LUS diagnoses (Table 3 and Table 4).

## 4. Discussion

In this retrospective pilot analysis, we describe for the first time seven CPUS phenotypes of acute respiratory distress in term and late preterm infants. We found that TTN, intended as wet lung, may account for a small part of the patients, with mild PPHN, mild MAS and mixed phenotypes being the possible causes of transient respiratory distress near birth for most of the infants included in the study. 

LUS has been used to study different populations of acute respiratory distress after birth [32,33]. The rates of acute neonatal respiratory disorders based on LUS that were reported in previous studies are slightly different compared to our findings [32,33]. However, our study was not powered to run epidemiological conclusions. Moreover, the inclusion criteria are different. We focused on patients with mild transient respiratory symptoms at admission that did not require invasive ventilation and were still on respiratory support at the time of CPUS. In addition, we combined for the first time the use of LUS with TnEcho, introducing a novel potential classification, based on a comprehensive cardiopulmonary approach. Therefore, a comparison with previous epidemiological LUS studies of neonatal acute respiratory distress is not feasible. 

Interestingly, we found a very high incidence of atelectasis at LUS in patients born through meconium-stained fluid. In our population, the incidence of meconium-stained amniotic fluid was quite high (37.5%) as compared to that of overall pregnancies, which is about 20% [34]. However, we only included infants with respiratory distress that needed respiratory support. It is described that MAS complicates 5% of deliveries with meconium-stained amniotic fluid [35]. MAS is usually considered a differential diagnosis of TTN, based on the presence of radiographic findings with patchy changes due to meconium-induced atelectasis. Approximately 33–49.7% of MAS-diagnosed neonates require ventilator support, with a 5–12% mortality rate [36,37]. Studies investigating LUS findings in MAS included only sick patients and focused on the agreement between chest X-ray and LUS [38]. In a recent work, Chiruvolu and co-authors reported that approximately 50% of the infants born through meconium-stained amniotic fluid and admitted to the NICU suffer from TTN. The authors diagnosed TTN according to the usual nonspecific radiological and clinical criteria [39]. On the contrary, in our population, none of the infants suffering from mild respiratory symptoms and being born through meconium-stained fluid presented with LUS signs of increased lung fluid (the sonographic hallmark of TTN), suggesting a different diagnosis. Despite being clinically and radiologically diagnosed with TTN, most of these patients (80%) showed atelectasis at LUS, with or without increased PVR at TnECHO, suggesting that the respiratory distress could be explained by stained amniotic fluid inhalation at birth. The rest of the patients born through meconium-stained amniotic fluid (20%), showed normal LUS, with TnECHO findings of increased PVR, suggesting mild PPHN. Our results, if confirmed in larger prospective studies, may carry important epidemiological insights. The possibility of recognizing small meconium-induced atelectasis that cannot be visualized at chest X-ray may imply that the incidence of MAS based on LUS is significantly higher than currently reported. While we cannot ultimately rule out the possibility that the atelectasis are due to a lack of recruitment or lung aeration in the delivery room, the distribution and appearance of the atelectasis that we found resemble the typical LUS description of MAS [24,26]. In the only case with consolidation at LUS that did not have meconium-stained amniotic fluid, the amniotic fluid was blood-stained, which could be a cause for respiratory distress, as previously reported [40]. 

Another striking result is that 50% of the infants (12 patients) showed signs of increased PVR, suggesting mild PPHN. Out of these, eight patients (33.3% of the entire population) had some concurrent LUS signs of pulmonary involvement. On the contrary, 4 patients (16.6% of the entire population) had normal LUS and therefore a purely vascular physiopathology. PPHN results from the failure of relaxation of the pulmonary vasculature at birth, leading to the shunting of non-oxygenated blood from the pulmonary to the systemic circulation. PPHN is considered a serious condition that requires early intervention and treatment to prevent severe hypoxia and various short-term and long-term morbidities. The role of increased PVR in causing transient respiratory symptoms that may be mistakenly interpreted as TTN was not reported previously. In addition, we observed decreased mean systemic blood pressures in infants with raised PVR at one and five hours of life compared to those with normal PVR. This finding may have been caused by a transient impairment of the left ventricular preload secondary to lower pulmonary venous return and the right-to-left shunting across the PDA, which is often seen in PPHN patients [41]. Furthermore, the bulging of the interventricular septum to the left, as documented by the pathological eccentricity index in this population, may have played a negative role in the systolic function of the left ventricle. These findings underline the significant, yet not severe, impairment of hemodynamic transition from fetal-to-neonatal circulation in infants with mild respiratory symptoms that show increased PVR. Interestingly, we found a trend between increased PVR and MVM. A connection between the vascular phenotype of chronic lung disease of prematurity and MVM has been described [42,43]. The association between increased PVR at birth and placental malperfusion deserves a prospective evaluation.

The analysis of post-natal outcomes did not reveal other significant associations, possibly because the sample size was not powered to detect significant differences among the CPUS phenotypes. However, the data from this pilot study may assist further in larger studies, helping to choose the appropriate sample size and target the research questions.

TTN often requires admission to the NICU to provide respiratory support, causing considerable parental stress and contributing significantly to health care costs. In order to reduce the impact of this transient, yet very frequent condition, therapeutic options are still being evaluated. In agreement with previous reports [44,45], a recent double-blind randomized controlled trial (RCT) proved that the treatment with 0.1 mg/kg of inhaled salbutamol significantly decreased the TTN clinical score, oxygen demands, and duration of respiratory support as compared to to the control group [46]. However, there was no significant difference between the groups in terms of hospital stay. Given the supposed origin of TTN based on decreased lung fluid clearance, randomized control trials studying the efficacy of furosemide [47], or racemic epinephrine, were conducted [48], but showed no significant difference in the duration of tachypnea or length of hospital stay compared with controls [49].

The identification of CPUS phenotypes, providing specific physiopathological clues, may pave the way for a more targeted approach, potentially improving patient inclusion and result precision of eventual future trials.

Longitudinal studies have shown an association between TTN and the subsequent development of asthma [50,51]. The definition of a more specific pattern of lung impairment may help the evaluation of long-term outcomes.

With regard to the technique used to perform the LUS evaluation, there is no agreement on the optimal approach. From a prognostic point of view, it has been suggested that the evaluation of the posterior lung fields may not be necessary and therefore could be avoided [52,53]. Our proposed AND score is diagnostic rather than prognostic. The worst lung field is chosen to make the diagnosis. We found that in most cases there is high variability in the LUS appearance of the different lung fields. The posterior lung fields were often more severely affected than the anterior ones and they were therefore chosen as the major diagnostic clue. From a diagnostic point of view, the evaluation of all lung fields seems crucial. Moreover, the possibility of LUS-guided postural recruitment may be missed if the posterior lungs are not examined [18,54].

Besides the retrospective nature of the analysis, our study has limitations. First, the CPUS examination was not performed at the same time-point for every infant, as it depended on the presence of a neonatologist trained in CPUS. Moreover in five infants (20.8%), LUS and TnECHO were performed at different hours of life (with a maximum time difference of 10 h), because the personnel trained in both techniques was not available. Moreover, we missed those infants (12 patients) that did not need non-invasive ventilation for more than a few hours as the mean hours of life at CPUS evaluation were 17+/−9 and the patients needed to be on respiratory support at the time of CPUS. Furthermore, due to the unavailability of personnel trained in CPUS, more than half of infants with suitable clinical characteristics were excluded, not receiving full ultrasound assessment on the first day of life. Furthermore, the CPUS follow-up was not standardized, so, we cannot provide data in this regard.

## 5. Conclusions

Our CPUS findings suggest that defining any mild respiratory distress such as TTN as related to delayed clearance of fetal lung fluid may be too simplistic. CPUS evaluation in patients with acute respiratory distress may improve diagnostic precision. A better understanding of the transient neonatal respiratory may assist communication between the team and the family, eventually reducing parental stress. The classification of pathophysiological phenotypes may also carry important epidemiological and experimental consequences.

## Figures and Tables

**Figure 1 children-10-00289-f001:**
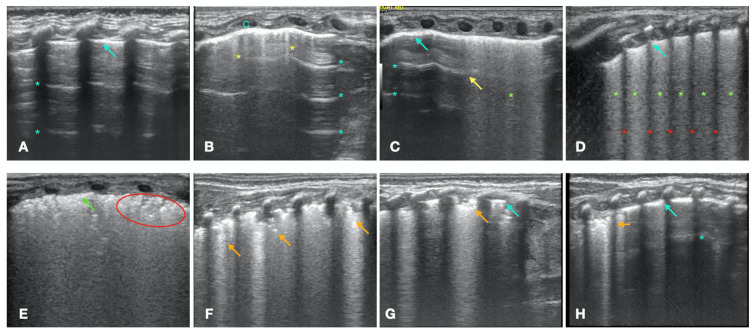
Lung ultrasound (LUS) possible findings. (**A**–**B**): *Normal lung:* A-lines (light-blue asterisks) characterizing the A-Pattern, with a normal, thin pleural line (light-blue arrow) (**A**); Sparse/isolated B-line(s) (yellow asterisks) characterizing the B1-Pattern; A-lines are present (light-blue asterisks), pleural line is normal (light-blue arrow) (**B**). (**C**,**D**) * Interstitial disease:* double lung point (yellow arrow), A-lines (light-blue asterisks) are present in the upper part, in the lower part B-lines are confluent (green asterisks), the pleural line is normal (light-blue arrow) (**C**); B2-Pattern, with confluent B-lines (green asterisks), the acustic shadows of the ribs is preserved (red asterisks), pleural line is normal (light-blue arrow) (**D**). (**E**): *RDS:* white lung with the disappearance of the acoustic echo of the ribs, thickened, irregular pleural line (green arrow) without spared areas, ground-glass sign (red circle). (**F**–**H**): *Irregular atelectasis* (orange arrows); multiple lung consolidation with irregular margins (**F**); basal lung consolidation with irregular margins, spared areas with normal, thin pleural line (light-blue arrow) (**G**); apical lung consolidation with irregular margins and spared areas with normal, thin pleural line (light-blue arrow) (**H**).

**Figure 2 children-10-00289-f002:**
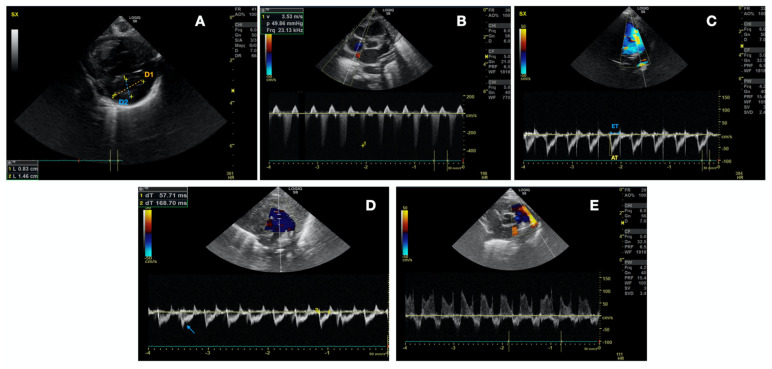
Echo findings in the case of increased vascular resistance. (**A**) Tele-Systolic eccentricity index (EI) > 1.15, calculated as D1 (orange dotted line) divided by D2 (blue dotted line). (**B**) pulmonary artery systolic pressure (PAPs) systemic or supra-systemic with doppler measurement of tricuspid regurgitation and pulmonary regurgitation jet. (**C**) Pulmonary artery acceleration time (dotted yellow line) to right ventricular ejection time ratio (full blue line) (PAAT/RVET) < 0.3. (**D**) Notch on the pulmonary artery flow (blue arrow). (**E**) PDA with bidirectional pattern with right-to-left shunt more than 30% (continuous right-to-left shunt may be present too).

**Figure 3 children-10-00289-f003:**
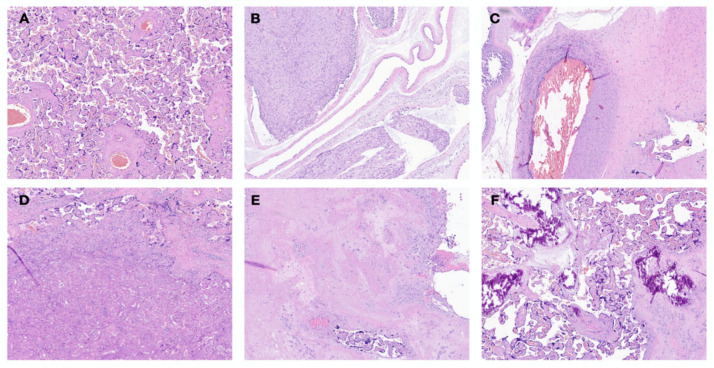
Main histopathological findings. Magnification 4x, hematoxylin-eosin (**A**) normal, (**B**) Chiorioamnionitis, (**C**) vasculitis, (**D**) ischemic necrosis, (**E**) fibrinoid necrosis, and (**F**) dystrophic calcifications.

**Table 1 children-10-00289-t001:** Lung Ultrasound (LUS) patterns and corresponding diagnosis.

LUS Pattern	LUS Diagnostic Score	LUS Diagnosis ^1^
A pattern	0	Normal Lung
B1 pattern	1	
Double lung point	2	Wet lung
B2 pattern	3	
White lung + pleural line abnormalities	4	RDS
+/− Ground-glass opacity sign or		
Snowflake sign		
Irregular atelectasis	5	Pulmonary consolidation

^1^ based on the higher score value in any lung field.

**Table 2 children-10-00289-t002:** Maternal and neonatal data.

Characteristics	*n* = 24
**Maternal data**	
Maternal diabetes, n (%)	2 (8.3%)
Maternal hypertensive disorders, n	0
Triple-I, n (%)	8 (33.3%)
IUGR, n (%)	1 (4.1%)
Fetal tachicardia, n (%)	6 (25%)
Maternal tachicardia, n (%)	1 (4.1 %)
Maternal fever, n (%)	3 (12.5%)
Intrapartum antibiotics, n (%)	8 (33.3%)
Chorioamnionitis, n (%)	9 (37.5%)
Placenta malperfusion, n (%)	12 (50%)
**Perinatal infant data**	
Gestational age (weeks), mean (SD)	39.2 (2.1)
Birth weight (grams), mean (SD)	3330.2 (566.6)
Need for any kind of resuscitation at birth, n (%)	11 (45.8%)
Need for endotracheal intubation at birth, n (%)	0
Apgar at 1 min, median (IQR]	9 (5–9)
Apgar at 5 min, median (IQR)	9 (8–10)
Stained amniotic fluid, n (%)	10 (41.6%)
Meconium stained amniotic fluid, n (%)	9 (37.5%)
Blood-stained amniotic fluid, n (%)	1 (4.1 %)
**Infant data related to NICU stay**	
Maximum CRP (mg/dL), mean (SD)	3.8 (5.1)
Positive blood culture, n (%)	1 (4.1 %)
Antibiotic therapy, n (%)	10 (41.6%)
Antibiotics duration (days), mean (SD)	2 (3.4)
Length of stay (days), mean (SD)	8.1 (3.7)
Duration of CPAP (hours), mean (SD)	28.5 (57)
Duration of HFNC (hours), mean (SD)	21.3 (51.2)
Overall duration of NRS (hours), mean (SD)	49 (46.7)
Duration of NRS/low-flow oxygen (hours), mean (SD)	62 (69)
Maximum FiO_2_, mean (SD)	0.36 (0.11)
Surfactant administration via INSURE, n (%)	4 (18.2)
Hours of life at CPUS, mean (SD)	13.9 (7.7)
FiO_2_ at CPUS, mean (SD)	0.35 (0.09)
**Discharge diagnosis**	
Transient tachypnea of the newborn, n (%)	22 (91.6%)
Respiratory distress syndrome, n (%)	1 (4.1%)
Meconium aspiration syndrome, n (%)	1 (4.1%)

Abbreviations: intrauterine growth restriction (IUGR), neonatal intensive care unit (NICU), C-reactive protein (CRP), continuous positive airway pressure (CPAP), high flow nasal cannula (HFNC), non invasive respiratory support (NRS)—intended as the sum of CPAP and HFNC, INSURE (Intubation-SURfactant-Extubation) cardiopulmonary ultrasound (CPUS). Results are reported as number (n) and percentages (%), mean and standard deviations (SD), or median and interquartile range [IQR], as appropriate.

**Table 3 children-10-00289-t003:** Acute neonatal distress (AND) phenotypes based on cardiopulmonary (CPUS) findings.

Lung Diagnosis	Raised PVR	AND Phenotypes
Normal Lung (*n* = 6, 25%)	No	Undefined (*n* = 2, 8.3%)
	Yes	Vascular (*n* = 4, 16.6%)
Wet lung (*n* = 8, 33.3%)	No	Wet lung (*n* = 4, 16.6%)
	Yes	Vascular wet lung (*n* = 4, 16.6%)
RDS (*n* = 2, 8.3%)	Yes	Vascular-RDS (*n* = 2, 8.3%)
Pulmonary consolidation (*n* = 8, 33.3%)	No	Consolidation (*n* = 6, 25%)
	Yes	Vascular-consolidation (*n* = 2, 8.3%)

Abbreviations: PVR = pulmonary vascular resistance, RDS = respiratory distress syndrome.

**Table 4 children-10-00289-t004:** Characteristics of infants classified according to cardiopulmonary (CPUS) phenotyping.

	Undefined (AND-u)	Vascular (AND-v)	Wet Lung (AND-w)	Vascular-Wet Lung (AND-vw)	Vascular-RDS	Consolidation (AND-c)	Vascular-Consolidation (AND-vc)	*p*-Value
	(*n* = 2)	(*n* = 4)	(*n* = 4)	(*n* = 4)	(*n* = 2)	(*n* = 6)	(*n* = 2)	
Gestational age (weeks), median (IQR)	40.4 (39.7–41.1)	41 (35.8–41.9)	37.5 (36–40.3)	36.8 (35.8–40-4)	37.9 (35.7–40.1)	40.2 (38.6–41)	40.3 (39.41.7)	0.45
Birth weight (grams), median (IQR)	3695 (3520–3870)	3392 (2772–3695)	3587 (3031–4287)	2812 (2715–3205)	3175 (2100–4250)	3405 (3078–3537)	3515 (3175–3855)	0.72
Hour of life at echocardiography (h), median (IQR)	17.5 (11–24)	18.5 (9.75–21.25)	10.5 (5.5–20)	9 (5.25–18)	24 (24–24)	14 (5.25–22-5)	6 (2–10)	0.28
Ejection fraction (%), median (IQR)	64.5 (60–69)	64 (60–65)	64 (60.75–65)	63 (58.5–66)	61.5 (58–64)	65 (58.75–72)	65 (65–65)	0.95
PAAT/RVET, median (IQR)	0.45 (0.44–0.46)	0.31 (0.22–0.46)	0.41 (0.32–0.52)	0.23 (0.17–0.40)	0.37 (0.34–0.40)	0.45 (0.38–0.48)	0.20 (0.20–0.20)	0.1
Mid-systolic notch of the flow across the pulmonary artery, n (%)	0	2 (50%)	1 (25%)	3 (75%)	0	0	2 (100%)	0.03
Tricuspid valve insufficiency, n (%)	0	3 (75%)	2 (50%)	2 (50%)	0	1 (16.6%)	2 (100%)	0.2
Trans-ductal flow pattern								0.22
R-to-L > 30% n (%)	0	2 (50%)	1 (25%)	1 (25%)	2 (100%)	4 (66.7%)	2 (100%)	
Growing, n (%)	0	2 (50%)	1 (25%)	3 (75%)	0	0	0	
Pulsatile, n (%)	0	0	1 (25%)	0	0	0	0	
Restrictive, n (%)	1 (50%)	0	0	0	0	1 (16.7%)	0	
No-PDA, n (%)	1 (50%)	0	1 (25%)	0	0	1 (16.7%)	0	
Trans-ductal peak systolic velocity (m/s), median [IQR]	2 (2–2) †	0.9 (0.9–0.9) ¶	1 (0.8–1.2)	1 (0.99–1.37)	1.05 (1–1.1)	1.8 (1.65–2.35) *,¶	0.8 (0.7–0.9) *,†	0.049
Tele-systolic eccentricity index, median (IQR)	0.97 (0.95–1) †,*,¶	1.63 (1.51–1.73) †,‡	1.1 (0.92–1.65) §	1.45 (1.33–1.87)	1.65 (1.5–1.7) §	1.18 (0.97-1.4) ‡	1.62 (1.55–1.7) *	0.045
Hour of life at lung ultrasound (h), median (IQR)	17.5 (11–24)	18.5 (9.75–21.25)	14.5 (5.75–21.75)	9 (5.2–18)	23.5 (23,24)	15 (5.25–25.75)	13 (2–24)	0.69
Pleural line								0.08
Normal, n (%)	1 (50%)	3 (75%)	4 (100%)	3 (75%)	0	2 (33.3%)	0	
Thickened, n (%)	1 (50%)	1 (25%)	0	1 (25%)	2 (100%)	2 (33.3%)	0	
Irregular, n (%)	0	0	0	0	0	2 (33.3%)	2 (100%)	
Focal subpleuric micriatelectasis, n (%)	0	0 †	0 *	1 (25%)	0	6 (100%) *,†	2 (100%)	<0.001
Anterior right AND score, median (IQR)	0.5 (0–1)	0 (0–0)	0.5 (0–1.75)	1.5 (0.25–4.25)	3.5 (3–4)	1 (0–3)	0.5 (0–1)	0.14
Anterior left AND score, median (IQR)	0.5 (0–1)	0 (0–0)	0.5 (0–2.5)	2 (0.25–4.5)	3.5 (3–4)	1 (0–3)	0.5 (0–1)	0.15
Posterior right AND score, median (IQR)	1.5 (0–3)	0 (0–0)	3 (3–3)	1.5 (0.25–2.75)	4.5 (4–5)	4 (2.25–5)	2.5 (0–5)	0.06
Posterior left AND score, median (IQR)	1.5 (0–3)	0 (0–0) *,†,⁑	3 (3–3) ¶	2 (0.5–2.75) ‡	4.5 (4–5) †	5 (3–5) *	5 (5–5) ‡,⁑	0.004

Abbreviations: PAAT/RVET = Ratio between pulmonary artery acceleration time and right ventricular ejection time; PDA = patent ductus arteriosus; AND = Acute Neonatal Distress, right to left (R-to-L). Symbols indicate *p* < 0.05 from the post-hoc analysis. ‡,†,*,¶,§,⁑: symbols represent significance among groups. Two groups with the same symbol are significant among each other.

## Data Availability

The data that support the findings of this study are available on request from the corresponding authors, C.B, G.G, M.P.

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
