# Peer review of "Cardiopulmonary Ultrasound Patterns of Transient Acute Respiratory Distress of the Newborn: A Retrospective Pilot Study"

_children, 2023, doi:10.3390/children10020289_

Round 1

Reviewer 1 Report

Dear editor, I read and reviewed carefully the paper from Pierro et al titled “Cardiopulmonary ultrasound assessment and phenotyping of 2 transient tachypnea of the newborn: a retrospective pilot study”. The work is a retrospective pilot study trying to identify cardiopulmonary ultrasound (CPUS) patterns in term and late preterm infants suffering from respiratory distress and requiring non-invasive respiratory support in the Neonatal and Pediatric Intensive Care Unit of the Bufalini Hospital in Cesena, Italy.

The idea is original, and the preliminary data have the potential to be the base of a more comprehensive and exhaustive study. However, there are a number of issues and concerns outlined below that I have with the paper and that need to be addressed before it can be accepted.

A general suggestion to the authors is to be more precise and clear because the paper is difficult to follow sometimes and some mistakes could have been easily avoided if the paper was written with more accuracy and attention.

Major points:

1.     Authors session: 

·      number 3 (that is associated with 3 authors) has just one email (that is not supposed to be there and anyway it is for just one author when the number 3 is associated with 3 authors) and one university’s name. 

·      some of the departments names are in Italian, some in English (for instance, 1 is in Italian but 5 is in English; correspondent author has her affiliations in Italian in number 1 but then at the end are in English). As well, please be consistent and more accurate.

2.     Material and methods:

·      Line 68: what are the exact inclusion criteria? infant with respiratory symptoms  ( and therefore which ones?) admitted in the NICU for no invasive respiratory support? Or infants with respiratory symptoms admitted in the NICU with or without respiratory support? or infants admitted with the diagnosis of TTN ( even tough TTN can be diagnosed only at discharge since respiratory distress can be caused by different entities?) Please be more specific cause it is confusing especially cause throughout the paper it looks that the inclusion criteria changed.

·      Line 71: what do the authors means with patients symptomatic? What was the definition? Some degree of Fio2 requirements? Or some degree of work of breathing and in that case, defined by which clinical signs?

·      Line77-82: please double check the informations presented in the table and describe all of them in the order they are presented in the table.

·      Line 77-82: the Fio2 and the HOL when the US was done need to be reported in this table as well as the discharge diagnosis. Respiratory distress in late preterm newborns not only can be secondary to different pathophysiological processes but also is an evolving process over time that for sure present different presentations  and images at the Ultrasound.

·      Line 93: Was the US findings/images reviewed by only one US experts? Shouldn’t be done by at least 2 people to have a more accurate and unbiased diagnosis?

·      Line 100-105: The different type of lines should be described earlier than the phenotypes as well as there should be arrows in the different pictures to help the reader to understanding what they are supposed to look at.

·      Line 100 and Line 103: it is confusing: at first the authors are talking about 5 types of lung patterns but then in the table they show 6 as well in the discussions line 291 they talk about 6 different patterns. Please clarify.

3.     Statistical analysis

·      Line 207: please correct with “univariate analyses were used to evaluate respiratory phenotype (do the author means the AND phenotypes? Or what else? ) with respect to the maternal history, perinatal characteristics, duration of ventilation, oxygen treatment and hospital length.

·      Line 211 and 212: The methods describe the use of means, medians, interquartile ranges, proportions and 95% confidence intervals but in the tables is not clear when they present these data. Please identify in the table which data you are presenting.

4.     Results:

·      Line 223: here the authors talk about patients more than 33 weeks but in material and methods are describing the population as older than 34 weeks (line 68). So what was the gestational age cut off? Please be consistent

·      Line 224: how many charts were consulted? And how were the consulted charts screened? identified?

·      Line 228: It is not clear what the authors means with no invasive ventilation: CPAP and HFNC are no invasive ventilation so maybe is NIPPV ( PEEP with BUR?)? If for No invasive ventilation they means the pool of patients that required either CPAP or HFNC then they should be more specific and the data should reflect this.

·      Table 2: it is not clear how the data are presented. First, they should be ordered based on maternal history, perinatal events, infant characteristics, management, discharge diagnosis etc etc. also the title should not be neonatal characteristic since there are also maternal data. Then ( as commented before) where the mean is presented? where the median, the interquartile ranges, proportions and 95% confidence intervals? Please identify the data and make them clearer to read.

·      Table 2:  What is between parenthesis in the data like apgar score or duration of HFNC? What is the point to show mean apgar score for example? Would it be more informative the mode in this case the median, the minimum and  maximum between parenthesis?

·      Table2: the table needs to include the discharge diagnosis that are in the figure legend as well as Fio2 requirement, use of surfactant, needs of intubation.

·      Line 233-242: please be consistent with the presentation of the data: either are presented as number or as percentage or as both; either the data are presented based on AND phenotype or discharge diagnosis. Also, it should be easier to follow the presentation of the results based on the table because as per now it is very difficult to follow. 

·      Line 243: AND phenotypes should be also described in table 1 to make it clearer. For sure the authors are well familiar with their description of AND phenotypes and LUS phenotypes so it is easier for them to understand what they are describing; but it should be clearer for the readers without having to go back and forth between tables.  A second table with the data presented clearly can help.

·      Line 261: in the description of the AND phenotypes, there is suddenly a diagnosis of RDS that I find difficult to justify in that position and understand how a LUS could possibly make a diagnosis of RDS. Please clarify. Also, if we consider the undefine phenotype, there are 6 patterns plus one undefined. This is also confusing.

·      Table 3: the phenotypes in the table should be organized the way that they are presented in the text so the undefined phenotype in the table should be last. Is the vascular RDS phenotype the RDS phenotype in line 261? Or the authors are talking about something else?

·      Please discuss the fact that LU was done at different time than tnECHo in the discussion session. Could this affect the classification for example since done at different time?

·      Line 278: the authors state: 2 showed increased PVR but then the following sentence they state that 3 had normal LUS and increase PVR. So how many patients had increased PVR? 2? 3? 5? In any case, which type of phenotype presented these 2infants with increased PVR? Cause the other 3 had a AND-v phenotype. 

·      Line 280: there was a trend towards an increased incidence of MVM compared to what? It is not clear.

·      In the supplementary data there are information regarding 4 intubated newborns. Why these patients were not excluded from the studied population since they required more than NIV? In any case, these 4 infants should be discussed in the discussion since the whole paper is about LUS in TTN. Or the authors want to talk more broadly about infants that require NICU admission for increased work of breathing? 

·      The mean hours on NIV in the suppl data under the “ undefined phenotype” catetfory is 133h but in the table 2 the total mean is 56h. Can the analysis be checked again? Or can these data be clarified?

5.     Discussion:

·      Line 314: I am not understanding why there is a citation at the end of the sentence if the authors are describing their population. Please clarify. Also, is it possible that the authors missed atelectasis in those infants with history of meconium stained fluid without aspiration phenotype and only with increased PVR? 

·      Line 315: no infant out of the ones with h/o meconium stained fluid? It is not clear.

·      Line 319-320: the authors are talking about inconsistency with the paper from ChiruvolU ( please correct in the text where it is mentioned at ChiruvolO) et al but it is not clear what is the inconsistency about. They state that 80% of the infants with h/o meconium stained fluid showed signs of MAS in their study but that Chiruvoly fount 50% of the infant with h/o meconium stained fluid had TTN. So are they comparing MAS vs TTN? Please clarify in the text.

·      Line 320: the authors can not make such conclusions out of a pilot study with sometimes just 2 patients for group. Please either remove this sentence or clarify that these are preliminary data from a pilot study.

·      Line 325: MAS is in the differential for respiratory distress not for TTN cause TTN has usually a different radiographic picture and clinical presentation (milder than MAS).

·      Line 328: the reference that the authors present ( 35: https://doi.org/10.1203/01.pdr.0000214685.31232.6a) is talking about the contribution of pancreatic phospholipase A2 to lung injury in meconium aspiration syndrome about  not about the MAS diagnosis and the frequency of MAS diagnosis in infants with meconium stained fluid. Since article 35 is being referenced to frequently, the authors should correct with the right reference and double check all the bibliography to be sure about other wrong references throughout the paper.

·      Line 338-340: this sentence is not clear. Is 50% of the infant admitted in the NICU presenting increased PVR or 16%? Again, if the criteria for inclusion in the study is need of respiratory support or increased work of breathing, this would include more infants than the ones with only the diagnosis of TTN. 

·      Line 367-369: now the authors are talking about mild phenotypes of MAS, PPHN and RDS without clarifying how they classify mild MAS or PPHN and how they extrapolate the diagnosis of mild MAS or PPHN based on the LUS data. Then, the following sentence is talking about mixed phenotype based on the LUS? The AND score? Clinical data?. Please revise and clarify.

·      Line 372: didn’t some of these patients required surfactant administration and therefore infant were intubated? So maybe this sentence should be changed to “ with mild respiratory symptoms at admission”?

·      Line 389: I never heard of drugs to treat TTN. Please clarify what do you mean with that sentence cause, as mentioned previously by the authors, there are not recommendation for drug treatment in TTN. Also the sentence needs to finish with a point.

·      in December 2022, while this paper was in review, a new manuscript on LUS in TTN was published in journal of pediatrics by Pezza et al. please include it in your discussion.

6.     Conclusions: the conclusions are totally missing.

7.     Consent: how the written consent was requested if the IRB was approved on October 7 2022 and the paper submitted in December? Were the families of the infants (whose charts were reviewed) contacted after discharge? Were or were not the LUS and TnECHo standard of care? If they were, why it was necessary a written consent for a retrospective study? If they were not, the consent should have been asked before doing anything and therefore the IRB was supposed to be approved before starting the study. Please clarify.

8.     Bibliography: 

·      many references are not properly cited. For example 14 should be : Liu J, Copetti R, Sorantin E, Lovrenski J, Rodriguez-Fanjul J, Kurepa D, Feng X, Cattaross L, Zhang H, Hwang M, Yeh TF, Lipener Y, Lodha A, Wang JQ, Cao HY, Hu CB, Lyu GR, Qiu XR, Jia LQ, Wang XM, Ren XL, Guo JY, Gao YQ, Li JJ, Liu Y, Fu W, Wang Y, Lu ZL, Wang HW, Shang LL. Protocol and Guidelines for Point-of-Care Lung Ultrasound in Diagnosing Neonatal Pulmonary Diseases Based on International Expert Consensus. J Vis Exp. 2019 Mar 6;(145). doi: 10.3791/58990. PMID: 30907892.

·      Or 15: Soldati G, Smargiassi A, Demi L, Inchingolo R. Artifactual Lung Ultrasonography: It Is a Matter of Traps, Order, and Disorder. Applied Sciences. 2020; 10(5):1570. https://doi.org/10.3390/app10051570

·      Also some of the papers cited are in the wrong place such as the number 33or the first author is misspelled in the text ( Chiruvolo instead of ChiruvolU). Please double check.

Minor points:

1.     Authors session: 

·      Some of the sentences finish with a comma, some with a point, some with a semicolon. please be consistent.

·      5 is Department of Pathology not pathological anatomy

2.     Material and Methods:

·       Line 72: the study was not done between July 2020 and July 2022 since the IRB was approved in october 2022. I would rephrase it as “data were collected from patients admitted in the NICU from July 2020 untill july2022”.

·       Line 83: prenatal complications should be changed in to “maternal history”. 

·       Line 102: please replace in-homogenous with dysomogenous.

·       Line 277: table 1 is describing the AND phenotypes not the PVR. Please double check which table do you want to cite. 

3.     Results:

·      Line 285: “In any case” should be changed to “ultimately” or something more scientific.  

4.     Discussion

·      Line 314: “the rest was…” not “ the rest were…”

·      Line 358-360: fascinating should be removed.

·      Line 363: please change analysis with “the data from this pilot study”. 

·      Line 393-394: “with regards to” and “to performed” is not really correct English. Either remove the S and change to “to perform” or rephrase using “ there is not agreement on the optimal approach regarding the technique for LUS evaluation”.

·      Line 396: it is a “proposed diagnostic score” since it is a pilot study.

Author Response

Dear editor, I read and reviewed carefully the paper from Pierro et al titled “Cardiopulmonary ultrasound assessment and phenotyping of 2 transient tachypnea of the newborn: a retrospective pilot study”. The work is a retrospective pilot study trying to identify cardiopulmonary ultrasound (CPUS) patterns in term and late preterm infants suffering from respiratory distress and requiring non-invasive respiratory support in the Neonatal and Pediatric Intensive Care Unit of the Bufalini Hospital in Cesena, Italy.

The idea is original, and the preliminary data have the potential to be the base of a more comprehensive and exhaustive study. However, there are a number of issues and concerns outlined below that I have with the paper and that need to be addressed before it can be accepted.

A general suggestion to the authors is to be more precise and clear because the paper is difficult to follow sometimes and some mistakes could have been easily avoided if the paper was written with more accuracy and attention.

Major points:

1.     Authors session: 

·      number 3 (that is associated with 3 authors) has just one email (that is not supposed to be there and anyway it is for just one author when the number 3 is associated with 3 authors) and one university’s name. 

·      some of the departments names are in Italian, some in English (for instance, 1 is in Italian but 5 is in English; correspondent author has her affiliations in Italian in number 1 but then at the end are in English). As well, please be consistent and more accurate.

A: We modified as suggested

2.     Material and methods:

  • Line 68: what are the exact inclusion criteria? infant with respiratory symptoms  ( and therefore which ones?) admitted in the NICU for no invasive respiratory support? Or infants with respiratory symptoms admitted in the NICU with or without respiratory support? or infants admitted with the diagnosis of TTN ( even tough TTN can be diagnosed only at discharge since respiratory distress can be caused by different entities?) Please be more specific cause it is confusing especially cause throughout the paper it looks that the inclusion criteria changed.

 A: Thanks for this thoughtful comment. The inclusion criteria were: GA>= 34,  need for non-invasive respiratory support, CPUS evaluation in the first 24 hours of life (while still on non invasive respiratory support). We modified the methods section to improve clarity and precision. 

  •  Line 71: what do the authors means with patients symptomatic? What was the definition? Some degree of Fio2 requirements? Or some degree of work of breathing and in that case, defined by which clinical signs

A: Thanks for this comment. By “still symptomatic” we meant that the patients were still requiring non invasive respiratory support when we performed the CPUS examination. We modified the text to avoid confusion

  • Line77-82: please double check the informations presented in the table and describe all of them in the order they are presented in the table.

A: We modified the disposition of different variables in the table, according to the suggestion

  •   Line 77-82: the Fio2 and the HOL when the US was done need to be reported in this table as well as the discharge diagnosis. Respiratory distress in late preterm newborns not only can be secondary to different pathophysiological processes but also is an evolving process over time that for sure present different presentations and images at the Ultrasound.

A: We included the suggested variables in the data collection and table, improving the patients’ characterization.

  • Line 93: Was the US findings/images reviewed by only one US experts? Shouldn’t be done by at least 2 people to have a more accurate and unbiased diagnosis?

A: Thanks for underlying this important point. The images were evaluated in duplicate, using Cohen’s kappa to report the chance of agreement between operators. We improved this part at the end of the methods (study design section) and in the results.

  • Line 100-105: The different type of lines should be described earlier than the phenotypes as well as there should be arrows in the different pictures to help the reader to understanding what they are supposed to look at

A: We inserted a brief explanation of the LUS artifacts, adding arrows/asterisks to the pictures and explanations in the caption.

  • Line 100 and Line 103: it is confusing: at first the authors are talking about 5 types of lung patterns but then in the table they show 6 as well in the discussions line 291 they talk about 6 different patterns. Please clarify

A: Thanks for this accurate comment. We used the word phenotypes in two different parts (AND phenotypes and LUS phenotypes) and this may have brought some confusion. We changed the LUS phenotypes into LUS diagnosis and maintained the AND phenotypes. We changed the names of different LUS diagnosis as suggested by reviewer 2. We identified: 5 LUS patterns, that were scored from 0 to 5 (LUS diagnostic score), obtaining 4 LUS diagnosis (two patterns were included in the same LUS diagnosis). We explained this better in the methods.

3.     Statistical analysis

  •  Line 207: please correct with “univariate analyses were used to evaluate respiratory phenotype (do the author means the AND phenotypes? Or what else? ) with respect to the maternal history, perinatal characteristics, duration of ventilation, oxygen treatment and hospital length

A: We clarified as suggested. 

4.     Results

  •  Line 211 and 212: The methods describe the use of means, medians, interquartile ranges, proportions and 95% confidence intervals but in the tables is not clear when they present these data. Please identify in the table which data you are presenting

A: We added to each line the specific statistic parameter in the tables.

  •  Line 223: here the authors talk about patients more than 33 weeks but in material and methods are describing the population as older than 34 weeks (line 68). So what was the gestational age cut off? Please be consistent

A: In the material and methods we wrote “born at 34 weeks or older" and in the results we wrote “above 33 weeks”, so the GA cut-off is still the same (>=34 or >33 weeks). Using two ways of reporting GA could be confusing, so we modified the the result section as kindly suggested. 

·      Line 224: how many charts were consulted? And how were the consulted charts screened? identified?

A: We reported all the required data in the population flow chart.

·      Line 228: It is not clear what the authors means with no invasive ventilation: CPAP and HFNC are no invasive ventilation so maybe is NIPPV ( PEEP with BUR?)? If for No invasive ventilation they means the pool of patients that required either CPAP or HFNC then they should be more specific and the data should reflect this.

A: We intended the total amount of non-invasive ventilation that was required, as the sum of HFNC and CPAP. The infants included in the analysis did not receive noninvasive positive pressure ventilation (NIPPV). We used the term NIV in the broad way meaning “the provision of any ventilatory support through the patient's upper airway using a mask or similar device”, that includes CPAP and HFNC as well, not intending NIV as a synonyms for NIPPV. Since the definition of NIV is debated and the term NIV may be confusing, we changed it  into non invasive respiratory support (NRS).

·      Table 2: it is not clear how the data are presented. First, they should be ordered based on maternal history, perinatal events, infant characteristics, management, discharge diagnosis etc etc. also the title should not be neonatal characteristic since there are also maternal data. Then ( as commented before) where the mean is presented? where the median, the interquartile ranges, proportions and 95% confidence intervals? Please identify the data and make them clearer to read.

A: Thanks for this comment. We modified as suggested.

·      Table 2:  What is between parenthesis in the data like apgar score or duration of HFNC? What is the point to show mean apgar score for example? Would it be more informative the mode in this case the median, the minimum and  maximum between parenthesis?

A: Thanks for this comment. We modified as suggested.

·      Table2: the table needs to include the discharge diagnosis that are in the figure legend as well as Fio2 requirement, use of surfactant, needs of intubation.

A: Modified as suggested. None of the infants was intubated for mechanical ventilation as this was an exclusion criteria, so all the intubations were performed to administer surfactant through INSURE technique

·      Line 233-242: please be consistent with the presentation of the data: either are presented as number or as percentage or as both; either the data are presented based on AND phenotype or discharge diagnosis. Also, it should be easier to follow the presentation of the results based on the table because as per now it is very difficult to follow. 

A: We improved consistency in regard to this point.

·      Line 243: AND phenotypes should be also described in table 1 to make it clearer. For sure the authors are well familiar with their description of AND phenotypes and LUS phenotypes so it is easier for them to understand what they are describing; but it should be clearer for the readers without having to go back and forth between tables.  A second table with the data presented clearly can help.

A: We added a table that hopefully is going to help the reader to understand and follow results+discussion. Thanks for this thoughtful comment.

·      Line 261: in the description of the AND phenotypes, there is suddenly a diagnosis of RDS that I find difficult to justify in that position and understand how a LUS could possibly make a diagnosis of RDS. Please clarify. Also, if we consider the undefine phenotype, there are 6 patterns plus one undefined. This is also confusing.

A: The diagnosis of RDS by lung ultrasound was made in accordance with previous reports (https://www.nature.com/articles/s41390-018-0114-9 - https://www.frontiersin.org/articles/10.3389/fped.2022.864911/full), as it now well established that LUS improves the efficiency of chest X-ray in the identification of patients suffering from RDS, independently from the surfactant administration, which is an operator-depended choice. We described the LUS RDS diagnosis better in the method section. We hope that now the overall clarity of the classification is improved

·      Table 3: the phenotypes in the table should be organized the way that they are presented in the text so the undefined phenotype in the table should be last. Is the vascular RDS phenotype the RDS phenotype in line 261? Or the authors are talking about something else?

A: We reorganized as suggested and added a clarification in the RDS part.

·      Please discuss the fact that LU was done at different time than tnECHo in the discussion session. Could this affect the classification for example since done at different time?

A: In a limited number of patients (n=6) the two components of CPUS assessment were performed with different timing. However, the inclusion criteria were respected (clinical criteria + LUS and TnEcho in the first 24 hours of life), therefore we included those patients in the analysis. The timing of LUS and TnECHO was always within 10 hours. The reason for this discrepancy is that, in routine clinical practice, the personnel trained in LUS and/or TnEcho was not always available at the same time and not all the personnel is trained in both techniques, We added this as a limitation of the study. Ideally, in a prospective study that aims to confirm what we observed, the two components of the CPUS assessment should be performed at same time and potentially being repeated over time.

·      Line 278: the authors state: 2 showed increased PVR but then the following sentence they state that 3 had normal LUS and increase PVR. So how many patients had increased PVR? 2? 3? 5? In any case, which type of phenotype presented these 2infants with increased PVR? Cause the other 3 had a AND-v phenotype. 

A: We corrected the typo, sorry about that. We hope that Table 3 may improve clarity.

·      Line 280: there was a trend towards an increased incidence of MVM compared to what? It is not clear.

A: We better specified at the end of the sentence the two groups that were compared.

·      In the supplementary data there are information regarding 4 intubated newborns. Why these patients were not excluded from the studied population since they required more than NIV? In any case, these 4 infants should be discussed in the discussion since the whole paper is about LUS in TTN. Or the authors want to talk more broadly about infants that require NICU admission for increased work of breathing? 

A: Thanks for this comment. These babies received endotracheal surfactant, via INSURE technique, without receiving invasive ventilation. We included all the infants that were admitted to the NICU and had transient symptoms that did not require mechanical ventilation. Therefore, we included these infants in the study, as they did not meet any exclusion criteria. We amended the text of the introduction and method sections, to improve clarity about this point.

·      The mean hours on NIV in the suppl data under the “ undefined phenotype” catetfory is 133h but in the table 2 the total mean is 56h. Can the analysis be checked again? Or can these data be clarified?

A: We confirm the analysis. The particularly high value of NIV duration in the Undefined phenotype was due to an outlier, which required prolonged NIV (HFNC) for 237 hours. 

5.     Discussion:

·      Line 314: I am not understanding why there is a citation at the end of the sentence if the authors are describing their population. Please clarify. Also, is it possible that the authors missed atelectasis in those infants with history of meconium stained fluid without aspiration phenotype and only with increased PVR? 

Most of the infants with meconium stained amniotic fluid and mild transient  respiratory symptoms are classified as TTN (as per references in the text). We found that most of these patients are likely mild MAS, and this would be a much higher number than currently reported in patients with transient respiratory symptoms and meconium-stained amniotic fluid. However, not all the infants with meconium-stained amniotic fluid, inhaled meconium, as expected. We believe it is extremely difficult that a meconium atelectasis is removed within few hours from birth. The patients with meconium-stained amniotic fluid and no signs of atelectasis at LUS had increased vascular resistances that explained their respiratory distress and they are likely to be patients with mild PPHN. Of course our data and hypothesis need to be confirmed in prospective studies

We removed the citation from the sentence in lime 314.

·      Line 315: no infant out of the ones with h/o meconium stained fluid? It is not clear.

A: We modified to improve clarity.

·      Line 319-320: the authors are talking about inconsistency with the paper from ChiruvolU ( please correct in the text where it is mentioned at ChiruvolO) et al but it is not clear what is the inconsistency about. They state that 80% of the infants with h/o meconium stained fluid showed signs of MAS in their study but that Chiruvoly fount 50% of the infant with h/o meconium stained fluid had TTN. So are they comparing MAS vs TTN? Please clarify in the text.

A: We corrected the typo, thank you. We modified the discussion, hopefully improving clarity.

  •  Line 320: the authors can not make such conclusions out of a pilot study with sometimes just 2 patients for group. Please either remove this sentence or clarify that these are preliminary data from a pilot study.

A: We changed the sentence as requested

·      Line 325: MAS is in the differential for respiratory distress not for TTN cause TTN has usually a different radiographic picture and clinical presentation (milder than MAS).

A: The results of our pilot study suggest that CPUS could modify the diagnosis in late-preterm/term infants suffering from mild-to-moderate respiratory distress, that would be generically classified as TTN. Indeed, it is possible that mild MAS is more common than previously thought. Future prospective studies will have to eventually confirm this assumption. Hence, we offer new insight into the MAS diagnosis, which apparently is not confined to those cases with specific radiographic pictures and moderate-to-severe clinical presentations.

·      Line 328: the reference that the authors present ( 35: https://doi.org 10.1203/01.pdr.0000214685.31232.6a) is talking about the contribution of pancreatic phospholipase A2 to lung injury in meconium aspiration syndrome about  not about the MAS diagnosis and the frequency of MAS diagnosis in infants with meconium stained fluid. Since article 35 is being referenced to frequently, the authors should correct with the right reference and double check all the bibliography to be sure about other wrong references throughout the paper.

A: Thanks for this comment. We fixed these references.

·      Line 338-340: this sentence is not clear. Is 50% of the infant admitted in the NICU presenting increased PVR or 16%? Again, if the criteria for inclusion in the study is need of respiratory support or increased work of breathing, this would include more infants than the ones with only the diagnosis of TTN. 

A: We modified the manuscript to improve the clarity of this point. 50% of infants (12 patients) had increased PVR. Out of these, 8 patients (33.3% of the entire population) had some concurrent lung parenchymal issue detected with LUS, while 4 patients (16.6%  of the entire population) had normal LUS and therefore a purely vascular physiopathology.

·      Line 367-369: now the authors are talking about mild phenotypes of MAS, PPHN and RDS without clarifying how they classify mild MAS or PPHN and how they extrapolate the diagnosis of mild MAS or PPHN based on the LUS data. Then, the following sentence is talking about mixed phenotype based on the LUS? The AND score? Clinical data?. Please revise and clarify.

A: We amended this part as suggested. The proposed diagnosis of PPHN was based on the presence of pure vascular component without parenchymal involvement (AND-v as classified in the results). The proposed diagnosis of MAS was based on the presence of a compatible history (stained amniotic fluid) and typical LUS pattern (characterized by the presence of lung consolidations with irregular margins, along with few spared areas – AND-c as classified in the results).

·      Line 372: didn’t some of these patients required surfactant administration and therefore infant were intubated? So maybe this sentence should be changed to “ with mild respiratory symptoms at admission”?

We amended as suggested.

·      Line 389: I never heard of drugs to treat TTN. Please clarify what do you mean with that sentence cause, as mentioned previously by the authors, there are not recommendation for drug treatment in TTN. Also the sentence needs to finish with a point.

A: We modified the sentence to ameliorate the clarity. There are trials on the use of salbutamol for TTN and also Cochrane reviews on this topic. They did not prove any efficacy.  Involving CPUS assessment in the patient selection to RCTs of drugs for TTN/patients with transient respiratory symptoms could assist trial design and therefore avoid exposure to useless interventions

·      in December 2022, while this paper was in review, a new manuscript on LUS in TTN was published in journal of pediatrics by Pezza et al. please include it in your discussion.

A: Thank you for the suggestion. We read with great interest the paper from Pezza et al., which showed that patients with RDS have worse lung aeration (based on a semi-quantitative lung ultrasound score) and oxygenation compared with those with TTN. Anyway, the study population was different compared to our pilot-study, as Pezza et al. enrolled very preterm and moderate preterm infants (instead of late preterm and term infants). This key difference and the semi-quantitative lung ultrasound score instead of a qualitative assessment that was proposed in our work make these two studies difficult to compare. Furthermore, important population characteristics, such as the rate of stained amniotic fluid, were not reported in the work of Pezza et al, and infants with PPHN were excluded from their analysis, but the diagnostic criteria were not explained, and the author did not clarify if all the infants systematically underwent echocardiography. We believe that there is very limited comparability of these two studies.

6.     Conclusions: the conclusions are totally missing.

Amended, thank you. Sorry about that.

7.     Consent: how the written consent was requested if the IRB was approved on October 7 2022 and the paper submitted in December? Were the families of the infants (whose charts were reviewed) contacted after discharge? Were or were not the LUS and TnECHo standard of care? If they were, why it was necessary a written consent for a retrospective study? If they were not, the consent should have been asked before doing anything and therefore the IRB was supposed to be approved before starting the study. Please clarify.

A: CPUS is routinely performed in our unit and it is provided without consent needed, in accordance with the availability of trained personnel. The written consent to retrieve data for retrospective studies was required by the local ethical committee, and was obtained.

8.     Bibliography: 

·      many references are not properly cited. For example 14 should be : Liu J, Copetti R, Sorantin E, Lovrenski J, Rodriguez-Fanjul J, Kurepa D, Feng X, Cattaross L, Zhang H, Hwang M, Yeh TF, Lipener Y, Lodha A, Wang JQ, Cao HY, Hu CB, Lyu GR, Qiu XR, Jia LQ, Wang XM, Ren XL, Guo JY, Gao YQ, Li JJ, Liu Y, Fu W, Wang Y, Lu ZL, Wang HW, Shang LL. Protocol and Guidelines for Point-of-Care Lung Ultrasound in Diagnosing Neonatal Pulmonary Diseases Based on International Expert Consensus. J Vis Exp. 2019 Mar 6;(145). doi: 10.3791/58990. PMID: 30907892.

·      Or 15: Soldati G, Smargiassi A, Demi L, Inchingolo R. Artifactual Lung Ultrasonography: It Is a Matter of Traps, Order, and Disorder. Applied Sciences. 2020; 10(5):1570. https://doi.org/10.3390/app10051570

·      Also some of the papers cited are in the wrong place such as the number 33or the first author is misspelled in the text ( Chiruvolo instead of ChiruvolU). Please double check.

A: Thanks for the comment. We amended and double-checked as needed.

Minor points:

1.     Authors session: 

·      Some of the sentences finish with a comma, some with a point, some with a semicolon. please be consistent.

·      5 is Department of Pathology not pathological anatomy.

2.     Material and Methods:

·       Line 72: the study was not done between July 2020 and July 2022 since the IRB was approved in october 2022. I would rephrase it as “data were collected from patients admitted in the NICU from July 2020 untill july2022”.

·       Line 83: prenatal complications should be changed in to “maternal history”. 

·       Line 102: please replace in-homogenous with dysomogenous.

·       Line 277: table 1 is describing the AND phenotypes not the PVR. Please double check which table do you want to cite. 

3.     Results:

·      Line 285: “In any case” should be changed to “ultimately” or something more scientific.  

4.     Discussion

·      Line 314: “the rest was…” not “ the rest were…”

·      Line 358-360: fascinating should be removed.

·      Line 363: please change analysis with “the data from this pilot study”. 

·      Line 393-394: “with regards to” and “to performed” is not really correct English. Either remove the S and change to “to perform” or rephrase using “ there is not agreement on the optimal approach regarding the technique for LUS evaluation”.

·      Line 396: it is a “proposed diagnostic score” since it is a pilot study.

A: we amended as suggested all the minor points.

Reviewer 2 Report

First of all, my congratulations to the research team for the work. The integration of lung ultrasound and functional echocardiography are the fundamental pillars for the pathophysiological understanding of the newborn with respiratory distress.

I agree that the joint evaluation can improve the individualized diagnostic and therapeutic approach.

Some considerations:

1.     Perhaps the correct term is not "transient tachypnea phenotypes" but pulmonary ultrasound patterns and their pathophysiological correlation with cardiorespiratory transition or similar...

2.     Material and Methods

2.1.  Study Desing

The CPUS assessment must be performed at the same time in order to establish a correlation between the two probes. It is noteworthy (retrospective study limitation) the wide time frame to carry out these studies (24h), which makes the results difficult to compare. It is noteworthy that in the group with the vascular phenotype, the ultrasound study was performed (only the pulmonary study was mencionated) later (23-24 h). Is there any explanation?

2.2.  LUS:

The classification of the B2 pattern as interstitial is contradictory to the literature. Figure 1D represents an alveolar-interstitial pattern, confluent B lines without A lines. It is true that it is a posterior plane, which is more frequently affected by gravity, but the classification gives rise to error. In fact, the image in Figure E (white lung with thickened an irregular pleural line) is practically the same as Figure D (except that it is an anterior plane).

Thus, the classification and the figures that identify the interstitial (B2-pattern) and white lung+ thickened and irregular pleural line phenotypes are not clear. Besides Score 4, is not specific to RDS, this statement is inaccurate.

2.3.  Statistical Analysis

All images are evaluated in duplicate. The evaluators:

-       Are they blind to the clinical diagnosis and evolution of the patient?

-       A K index is mentioned to assess interobserver variability, but it is not mentioned for what test, for lung ultrasound, echocardiography or both? And what is the result?

An adequate K index in the assessment of the left ventricle telesystolic IE is surprising, when a high intra- and interobserver variability is known, even an expert.

3.     Results:

3.1.  Small sample, almost half of those excluded.

3.2.  AND phenotypes, perinatal features and outcome

Characteristics of infants: I think it would be interesting to include the variables associated with perinatal asphyxia as a pathophysiological factor that increases PVR, for example: ph Au, need for CPR. In fact, although the PCR variable is not significant (perhaps due to the size of the sample), phenotypes with Atelectasis do require more PCR. Have you considered the possibility of lack of recruitment or lung aeration rather than fluid aspiration as a pathophysiological factor of cardiorespiratory maladaptation?

What is the clinical evolution of the phenotypes with atelectasis? It seems interesting to analyze its evolution.

4.     Discussion

As mentioned in line 310: it is correct to speak of different echographic patterns (cardiorespiratory) in patients with symptoms of transient respiratory distress. Being incorrect, pathophysiologically speaking, to include all the clinical situations in the entity of TTN.

Finally, I agree on the importance of pathophysiologically distinguishing the different types of transient respiratory distress, and their impact on diagnosis and treatment. Insist on the concept that not all are TTN.

Joint cardiorespiratory evaluation with lung ultrasound and echocardiography are essential.

I totally agree with the need to assess the posterior field (line 396) to establish a diagnosis and pathophysiological approach.

Congratulations.

Author Response

Reviewer 2

First of all, my congratulations to the research team for the work. The integration of lung ultrasound and functional echocardiography are the fundamental pillars for the pathophysiological understanding of the newborn with respiratory distress.

I agree that the joint evaluation can improve the individualized diagnostic and therapeutic approach.

Some considerations:

Comment 1

Perhaps the correct term is not "transient tachypnea phenotypes" but pulmonary ultrasound patterns and their pathophysiological correlation with cardiorespiratory transition or similar...

A: Thank you for this insight. We modified as kindly suggested.

Comment  2: Material and Methods. 2.1.  Study Desing

The CPUS assessment must be performed at the same time in order to establish a correlation between the two probes. It is noteworthy (retrospective study limitation) the wide time frame to carry out these studies (24h), which makes the results difficult to compare. It is noteworthy that in the group with the vascular phenotype, the ultrasound study was performed (only the pulmonary study was mencionated) later (23-24 h). Is there any explanation?

A:  Thanks for this comment. We added this part in the limitation section. In a limited number of patients (n=6) the two components of CPUS assessment were performed with different timing. The timing of LUS and TnECHO was always within 10 hours. Since the inclusion criteria were respected (clinical criteria + LUS and TnEcho in the first 24 hours of life), we included those patients in the analysis. The reason for this discrepancy is that, in routine clinical practice, the personnel trained in LUS and/or TnEcho was not always available at the same time and not all the personnel is trained in both techniques, We added this as a limitation of the study. Ideally, in a prospective study that aims to confirm what we observed, the two components of the CPUS assessment should be performed at same time and potentially being repeated over time. 

Comment  3: Material and Methods. 2.2.  LUS

The classification of the B2 pattern as interstitial is contradictory to the literature. Figure 1D represents an alveolar-interstitial pattern, confluent B lines without A lines. It is true that it is a posterior plane, which is more frequently affected by gravity, but the classification gives rise to error. In fact, the image in Figure E (white lung with thickened an irregular pleural line) is practically the same as Figure D (except that it is an anterior plane).

Thus, the classification and the figures that identify the interstitial (B2-pattern) and white lung+ thickened and irregular pleural line phenotypes are not clear. Besides Score 4, is not specific to RDS, this statement is inaccurate.

A: Thank you for the comment.  As suggested, we modified the LUS diagnosis, that could give rise to some confusion. The reason for the score was to evaluate the different lung regions from a statistic point of view, as we noticed that they often show different patterns among each other and make the diagnosis easier and more consistent among operators. We chose a more representative picture for RDS and we explained better the RDS features in the methods section. We hope that this part is now better explained. 

Comment  4: Material and Methods. 2.3.  Statistical Analysis

All images are evaluated in duplicate. The evaluators:

-       Are they blind to the clinical diagnosis and evolution of the patient?

-       A K index is mentioned to assess interobserver variability, but it is not mentioned for what test, for lung ultrasound, echocardiography or both? And what is the result?

An adequate K index in the assessment of the left ventricle telesystolic IE is surprising, when a high intra- and interobserver variability is known, even an expert.

A: Thanks for this comment. The second evaluator was blind to the clinical diagnosis and evolution of the patient, while this blinding process was impossible for the first operator, that was the clinician managing the patient and using CPUS at the bedside. We added the K indexes obtained from the analysis (for LUS and for TnEcho) in the Result section. We evaluated the K score in the assessment of “increased PVR yes/no” (based on the criteria and thresholds reported in the methods). This could explain the strikingly high degree of agreement among evaluators, impossible to reach in the case of continuous parameters.

Comment  5: Material and Methods.3.1.  

Small sample, almost half of those excluded.

A: We commented on this in the “limitations” sections. Unfortunately, half of the neonates with suitable clinical characteristics were excluded due to the unavailability of personnel trained in CPUS and a few had to be excluded because they were off respiratory support at the time of CPUS

Comment  6: Material and Methods. 3.2.  AND phenotypes, perinatal features and outcome

Characteristics of infants: I think it would be interesting to include the variables associated with perinatal asphyxia as a pathophysiological factor that increases PVR, for example: ph Au, need for CPR. In fact, although the PCR variable is not significant (perhaps due to the size of the sample), phenotypes with Atelectasis do require more PCR. Have you considered the possibility of lack of recruitment or lung aeration rather than fluid aspiration as a pathophysiological factor of cardiorespiratory maladaptation

What is the clinical evolution of the phenotypes with atelectasis? It seems interesting to analyze its evolution.

A: Thanks for these insightful comments and inputs for the discussion.

About the perinatal asphyxia, unfortunately, we did not collect the pH. Anyway, we reported the need for any resuscitation and we added the need for endotracheal intubation at birth (no infant required it). Furthermore, Apgar scores at 1 and 5 minutes were documented as well.

We added the the possibility of lack of recruitment as a possible cause of atelectasis, as we can’t ultimately know the nature of this ultrasound pattern with a 100%certanty. We commented on that in the discussion.

Clinical evolution is a fascinating matter. Thanks for this point. There are quite many factors playing a role in the clinical evolution (i.e., extension and the number of atelectasis, presence of increased PVR). Given the size of the sample size, we believe this aspect could be more accurately assessed in a prospective study.

Comment  7:  Discussion

As mentioned in line 310: it is correct to speak of different echographic patterns (cardiorespiratory) in patients with symptoms of transient respiratory distress. Being incorrect, pathophysiologically speaking, to include all the clinical situations in the entity of TTN.

A: We amended as suggested.

Comment  8:  Discussion

Finally, I agree on the importance of pathophysiologically distinguishing the different types of transient respiratory distress, and their impact on diagnosis and treatment. Insist on the concept that not all are TTN.

Joint cardiorespiratory evaluation with lung ultrasound and echocardiography are essential.

I totally agree with the need to assess the posterior field (line 396) to establish a diagnosis and pathophysiological approach.

Congratulations. 

A: Thank you very much

Reviewer 3 Report

Dear editor,

I would like to thank you for the opportunity to review the submitted manuscript titled: "Cardiopulmonary ultrasound assessment and phenotyping of transient tachypnea of the newborn: a retrospective pilot study".

I found this article interesting and easy to read. Pierro and colleagues have performed a retrospective observational study to analize the sonographic lung patterns in TTN and the presence of increased pulmonary vascular resistances in newborns at term or near term suffering from respiratory distress and needing respiratory support in the first 24 hours of life.

I have a few comments and suggestions to better help the readers understand the value of these results. Please see my comments below.

- Introduction:  No concerns.

 - Materials and Methods: well documented.

                 *Patients, Equipment, Study Design: no concerns.

                 * LUS data: authors use a cualitative score (AND score) to analize lung aireation. They consider 5 types of lung pattern scored from 0 (normal lung) to 5 (irregular atelectasis). Taking into account this score they identify 4 possible phenotypes: normal lung (score 0 and 1), interstitial disease (score 2 and 3), RDS (score 4) and pulmonary consolidation (score 5). In this point, I would suggest the authors to make a consideration as scores 4 and 5 are not ecographic patterns indicative of TTN.

                *Echocardiographic data: no concerns.

                 *Placental histology: I would like to ask authors if the placenta of all newborns included in the study were analized.

-         Statistical analysis: no concerns.

Results:

-           Line 231: as the study is based on newborn with diagnosis of transient tachypnea, patients with discharge diagnosis of RDS and MAS should be excluded?

-          Sonographic signs of TTN include double lung point and interstitial pattern in dependent areas with a thin pleural line. In fact, the presence of consolidations in lung ultrasound suggests against diagnosis of TTN (referencia) and supports other diagnosis such as RDS or MAS. I would suggest the authors to exclude these newborns, as their lung ultrasound does not support a diagnosis of TTN.

-          Table 2: a duration of oxygen administration mean of 59,73 hours is reported which is long for TTN, but maximum oxygen inspired fraction is not provided.

-          I wonder if the evolution of lung ultrasound could be provided. When was normalization of lung ultrasound achieved for each category?

Discussion: the major concern is the inclusion in the study of patients in which lung ultrasound shows consolidations because this finding rules out the diagnosis of TTN. This study may refer to newborns with mild symptoms of transient respiratory distress, term or near term, rather than classical diagnosis of TTN. Transient respiratory distress involves more mechanisms than insufficient reabsorption of lung fluid such as poor lung expansion at birth or increased RVP with normal lung ultrasound. In order to analyze this mechanisms, pulmonary and cardiac ultrasound assessment is very important, as the authors affirm in the discussion. Lung ultrasound assessment must include posterior areas, which is one of the strengths of this study.  

 Conclusions: no conclusions are provided.

Author Response

Dear editor,

I would like to thank you for the opportunity to review the submitted manuscript titled: "Cardiopulmonary ultrasound assessment and phenotyping of transient tachypnea of the newborn: a retrospective pilot study".

I found this article interesting and easy to read. Pierro and colleagues have performed a retrospective observational study to analize the sonographic lung patterns in TTN and the presence of increased pulmonary vascular resistances in newborns at term or near term suffering from respiratory distress and needing respiratory support in the first 24 hours of life.

I have a few comments and suggestions to better help the readers understand the value of these results. Please see my comments below.

 Comment 1 Materials and Methods. LUS data: 

Authors use a cualitative score (AND score) to analize lung aireation. They consider 5 types of lung pattern scored from 0 (normal lung) to 5 (irregular atelectasis). Taking into account this score they identify 4 possible phenotypes: normal lung (score 0 and 1), interstitial disease (score 2 and 3), RDS (score 4) and pulmonary consolidation (score 5). In this point, I would suggest the authors to make a consideration as scores 4 and 5 are not ecographic patterns indicative of TTN.

A: Thanks for this insightful comment. Indeed, an interesting result of this retrospective pilot study is that a notable part of patients clinically classified as TTN have an underlying cardiopulmonary physiopathology suggestive of different mechanism rather than accumulated interstitial fluid. We commented on that in the discussion. We also modified LUS diagnosis names to improve clarity.

Comment 2. Placental histology: 

I would like to ask authors if the placenta of all newborns included in the study were analized.

A: Yes, all the placentas were analyzed.

Comment  3.

Line 231 As the study is based on newborn with diagnosis of transient tachypnea, patients with discharge diagnosis of RDS and MAS should be excluded?

A: Thanks for this thoughtful comment. The inclusion criteria were: GA>= 34,  need for non-invasive respiratory support, CPUS evaluation in the first 24 hours of life (while still on non invasive respiratory support). Eventually, almost all the patients were discharged from the unit with a clinical diagnosis of TTN, as reported. It is fascinating that retrospective re-evaluation of the CPUS images lead to a different classification and sonographic diagnosis. We changed introduction, methods and discussion in order to improve clarity on this point.

Comment  4. 

Sonographic signs of TTN include double lung point and interstitial pattern in dependent areas with a thin pleural line. In fact, the presence of consolidations in lung ultrasound suggests against diagnosis of TTN (referencia) and supports other diagnosis such as RDS or MAS. I would suggest the authors to exclude these newborns, as their lung ultrasound does not support a diagnosis of TTN.

A: As discussed in the previous comment, the point we wanted to discuss with this study is the classification of late preterm and term infants with transient acute distress symptoms. We believe it is crucial to include in the analysis patients with clinical non-specific features suggestive of TTN (having transient self-limiting respiratory symptoms), but eventually re-classified in accordance with CPUS. We modified the text (introduction, methods, discussion) in order to make our aim and the consistent study design more clear.

Comment  5.

 Table 2: a duration of oxygen administration mean of 59,73 hours is reported which is long for TTN, but maximum oxygen inspired fraction is not provided.

A: We added this information to the table.

Comment  6

  I wonder if the evolution of lung ultrasound could be provided. When was normalization of lung ultrasound achieved for each category?

A: Unfortunately, the ultrasound follow-up was not standardly performed in every patient, and given the small sample size, we cannot carry an informative analysis out of the data available to us. We commented on this in the limitation section. A larger prospective study with systematic follow up is warranted.

Comment  7

Discussion: the major concern is the inclusion in the study of patients in which lung ultrasound shows consolidations because this finding rules out the diagnosis of TTN. This study may refer to newborns with mild symptoms of transient respiratory distress, term or near term, rather than classical diagnosis of TTN. Transient respiratory distress involves more mechanisms than insufficient reabsorption of lung fluid such as poor lung expansion at birth or increased RVP with normal lung ultrasound. In order to analyze this mechanisms, pulmonary and cardiac ultrasound assessment is very important, as the authors affirm in the discussion. Lung ultrasound assessment must include posterior areas, which is one of the strengths of this study.  

A: Thanks for this thoughtful comment. As we previously answered, it is crucial from our point of view to include patients with generic mild-to-moderate respiratory symptoms, showing the different physiopathological basis documented via comprehensive CPUS assessment. We believe this is one of the most notable results of our pilot study and take home messages.

Comment  8

 Conclusions: no conclusions are provided.

A: Fixed, as needed. Sorry about that.

Round 2

Reviewer 1 Report

all my questions were answered. 

Reviewer 3 Report

Dear editor, 

I believe the authors have successfully responded my previous concerns.

I agree with the authors that it would be very useful a larger prospective study for a better knowledge of this pathology. A systematic follow up of the lung ultrasound evolution could differenciate mild meconium aspirations from other conditions such as poor recruitment at birth, in which the normalization of lung ultrasound is achieved in less than 12-24 hours.   

Best regards.